# Stromal expression of hemopexin is associated with lymph-node metastasis in pancreatic ductal adenocarcinoma

Yukie Suzuki[1], Tatsuyuki Takadate[1]*, Masamichi Mizuma[1], Hiroki Shima[2], Takashi Suzuki[3], Tomoyoshi Tachibana[1], Mitsuhiro Shimura[1], Tatsuo Hata[1], Masahiro Iseki[1], Kei Kawaguchi[1], Takeshi Aoki[1], Hiroki Hayashi[1], Takanori Morikawa[1], Kei Nakagawa[1], Fuyuhiko Motoi[1], Takeshi Naitoh[1], Kazuhiko Igarashi[2], Michiaki Unno[1]

1 Department of Surgery, Tohoku University Graduate School of Medicine, Sendai, Miyagi, Japan,
2 Department of Biochemistry, Tohoku University Graduate School of Medicine, Sendai, Miyagi, Japan,
3 Department of Pathology and Histotechnology, Tohoku University Graduate School of Medicine, Sendai, Miyagi, Japan

* takadate@surg.med.tohoku.ac.jp

## Abstract

Pancreatic ductal adenocarcinoma is one of the most aggressive types of cancer. Certain proteins in the tumor microenvironment have attracted considerable attention owing to their association with tumor invasion and metastasis. Here, we used proteomics to identify proteins associated with lymph-node metastasis, which is one of the prognostic factors. We selected lymph node metastasis-positive and -negative patients (n = 5 each) who underwent pancreatectomy between 2005 and 2015 and subjected to comprehensive proteomic profiling of tumor stroma. A total of 490 proteins were detected by mass spectrometry. Software analysis revealed that nine of these proteins were differentially expressed between the two patient groups. We focused on hemopexin and ferritin light chain based on immunohistochemistry results. We assessed the clinicopathological data of 163 patients and found that hemopexin expression was associated with UICC N2 ($p = 0.0399$), lymph node ratio ($p = 0.0252$), venous invasion ($p = 0.0096$), and lymphatic invasion ($p = 0.0232$). Notably, *in vitro* assays showed that hemopexin promotes invasion of the pancreatic cancer cells. Our findings suggest that hemopexin is a lymph node metastasis-associated protein that could potentially serve as a useful therapeutic target or biomarker of pancreatic ductal adenocarcinoma.

## Introduction

Pancreatic ductal adenocarcinoma (PDAC) is one of the most aggressive types of cancer. The five-year survival rate from diagnosis is only 7% even after curative resection. Because of high recurrence rates, only 20% of the patients survive beyond five years [1–3]. For patients who have undergone PDAC resection, several prognostic characteristics, including tumor size, disease stage, histological grade, number of lymph-node (LN) metastases, R status, and adjuvant

**Data Availability Statement:** All relevant data are within the manuscript and its Supporting Information files.

**Funding:** This work was supported, in part, JSPS (Japan Society For The Promotion Of Science <URL>https://www.jsps.go.jp/j-grantsinaid/16_rule/rule_h24.html) KAKENHI Grant number 16K19914 and 19K18107. Tatsuyuki Takadate received the grant. The funder had no role in study design, data collection and analysis, decision to publish, or preparation of the manuscript. There was no additional external funding received for this study.

**Competing interests:** The authors have declared that no competing interests exist.

chemotherapy have been reported [4–7]. Notably, survival rate decreases with high LN-metastasis number–among patients with $\geq 4$ metastases, the five-year survival rate is 25% lower than that among patients with no metastasis [8]. Therefore, suppressing LN metastasis is critical for improving PDAC prognosis.

PDAC is histologically characterized by an abundant tumor stroma surrounding cancer cells. Cancer-associated fibroblasts (CAF), inflammatory cells, and immune cells such as lymphocytes and macrophages, and the extracellular matrix generally constitute the stromal tumor microenvironment [9–11]. It has been reported that interactions between the components of the tumor microenvironment and cancer cells promote cell proliferation, inflammation, and metastasis by activating various signaling pathways, including transforming growth factor β/SMAD and hepatocyte growth factor pathways [12,13]. Thus, in addition to cancer cells and these downstream signaling pathways, the tumor stroma has attracted attention as a new treatment target.

Recently, proteomics has been broadly used in clinical applications mainly to elucidate cellular functions and to identify biomarkers for malignant diseases. Specifically, proteomics performed using formalin-fixed paraffin-embedded (FFPE) tissues is a favorable method because of convenient sample preparation–surgical specimens are commonly preserved as FFPE tissues and typically archived together with corresponding clinical information. Previously, after pathological diagnosis, these specimens were used only as research samples for a few experimental methods such as immunohistochemistry, because the formalin-induced intra- or inter-molecular covalent crosslinking of amino acids made protein extraction challenging. However, large-scale proteomics using archival FFPE tissues was facilitated by advances in mass spectrometry (MS) and laser microdissection (LMD), which are effective tools for isolating target cells from heterogeneous tissues. By using these approaches, we previously successfully identified new biomarkers of pancreatic cancer, bile-duct cancer, and liver metastasis of pancreatic neuroendocrine neoplasm [14–16]. With increasing attention devoted to stromal functions, proteomics research has expanded to cover not only cancer cells but also the tumor microenvironment [17]. However, only a few studies have comprehensively analyzed stromal proteins [18–20].

Here, we aimed to identify the LN-metastasis-associated proteins in PDAC stroma by comparative proteomic profiling of specimens from patients with several LN metastases and those without any metastasis. To comprehensively investigate protein expression, we conducted retrospective proteomic analyses focused on the PDAC stroma by using liquid chromatography-tandem MS (LC-MS/MS) and FFPE specimens collected using LMD.

## Materials and methods

### Patients

Our study was approved by the Institutional Review Board of Tohoku University (Reference number: 2016-1-149). All methods were performed in accordance with the principles expressed in the Declaration of Helsinki. Written informed consent was obtained from all patients before surgery. All data were fully anonymized before we accessed them. Patients' medical records were accessed between April 2016 to September 2018.

Participants were selected from 355 pancreatic-cancer patients who underwent pancreatectomy between 2005 and 2015 at Tohoku University Hospital. Fig 1 shows the flowchart of patients whose specimens were subjected to LC-MS/MS. We selected 183 patients with histologically diagnosed invasive PDAC with no neoadjuvant therapy; 42 and 141 patients were LN-metastasis-negative (LN$^-$) and -positive (LN$^+$), respectively. We applied the following selection criteria: (1) tumor size, $\geq 20$ mm (pathological T$\geq$2, according to UICC 8$^{th}$ edition);

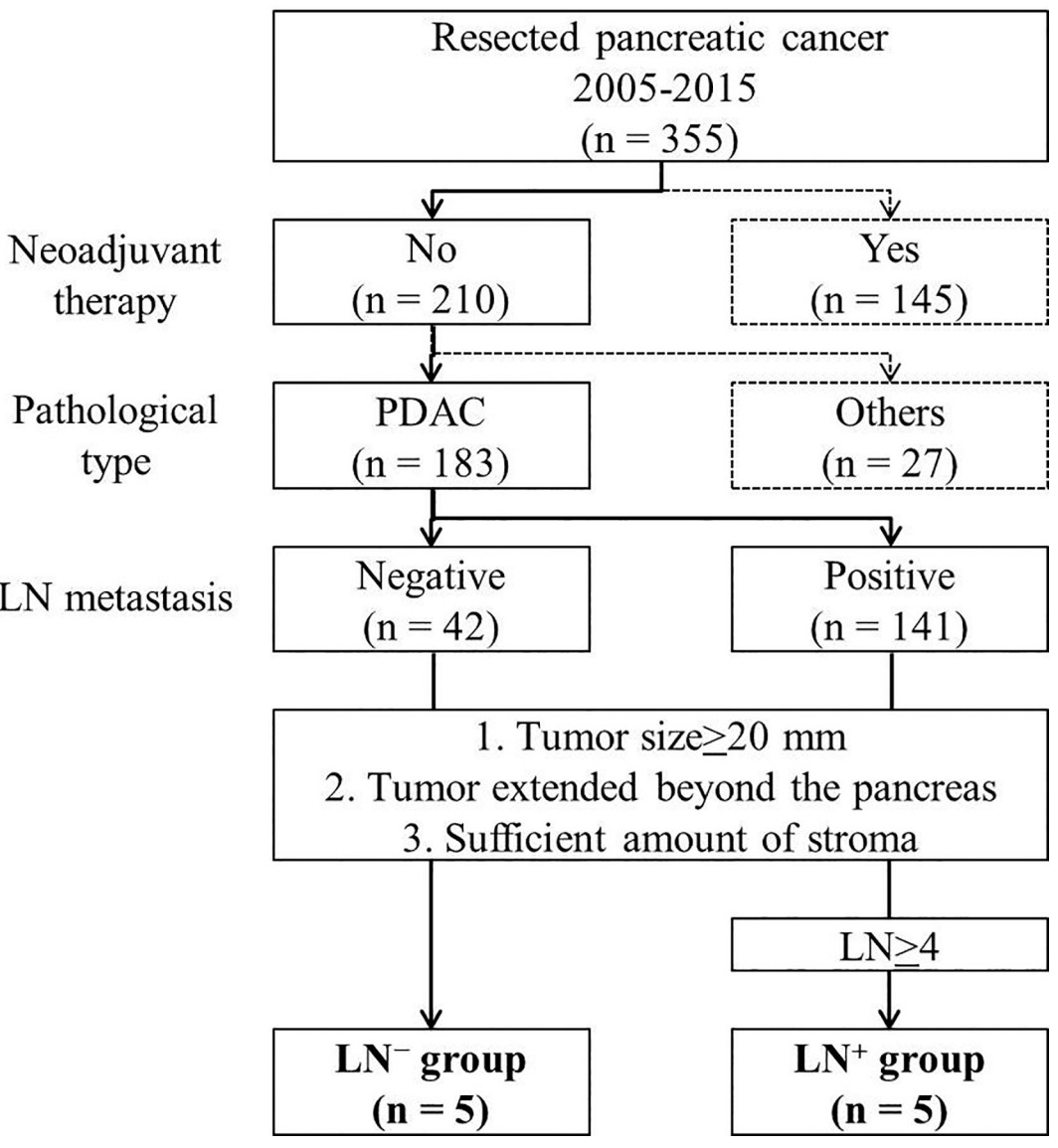

**Fig 1. Patient selection for specimen analysis by LC-MS/MS.** Study participants were selected from 355 pancreatic cancer patients who underwent pancreatectomy between 2005 and 2015 at Tohoku University Hospital. All patients in LN+ group were selected from patients with ≥4 LN metastases.

(2) tumor extended beyond pancreas; (3) sufficient amount of tumor stroma present for shotgun proteomic analysis. From the included patients, 10 were selected and further classified into two groups (n = 5 each) based on histopathological diagnosis: LN− group, no LN metastasis; LN+ group, ≥4 LN metastases. For clinicopathological analysis, we selected 163/183 patients with histologically diagnosed invasive PDAC; the patients had received no neoadjuvant therapy. We excluded the 10 patients whose specimens were selected for LC-MS/MS analysis and 10 whose specimens were unavailable.

## LMD of FFPE tissue sections and peptide extraction

Resected tumor tissues were fixed routinely in 10% neutral, buffered formalin at room temperature for 24–72 h and then embedded in paraffin for serial sectioning. We targeted only

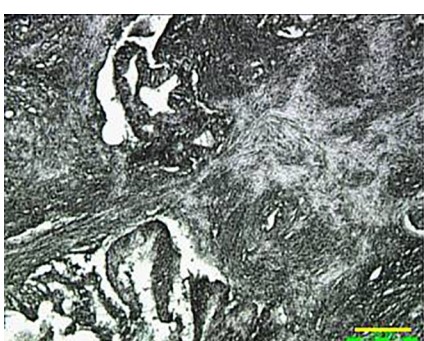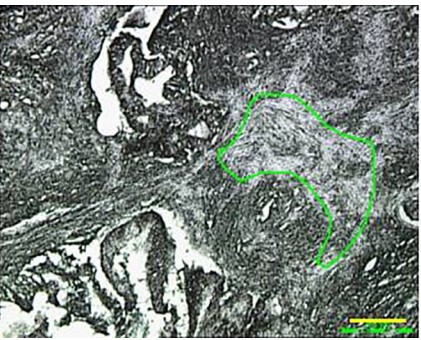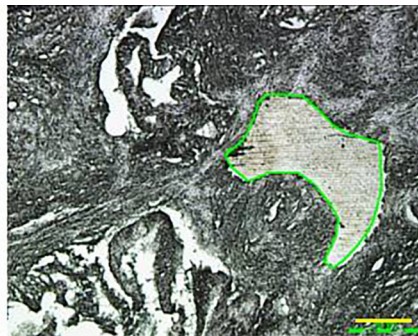

**Fig 2. Micrographs showing the PDAC stroma before and after microdissection.** LMD was used to dissect the stroma extending ≤2 mm away from cancer cells (outlined area). The targeted area (total 8 mm²) was selectively dissected from each sample. Magnification: 100×. Scale bar: 100 μm.

stromal areas of the tumor tissues, defined as tissue extending ≤2 mm away from cancer cells [21]. The targeted areas, which were confirmed in serial sections stained with hematoxylin and eosin, did not include cancer cells, islet cells, or vasculature. To extract proteins for shotgun proteomics, we used an LMD microscope, Leica LMD7500 (Leica Microsystems GmbH, Wetzlar, Germany). First, 10-μm-thick sections from the same FFPE block were applied to DIRECTOR™ slides (Expression Pathology Inc., Rockville, MD, USA), and sections were deparaffinized thrice (5 min each) in xylene, rehydrated in graded ethanol solutions and finally distilled water, and air-dried. From each patient sample, a stromal area was targeted and selectively microdissected. Stromal samples were collected from different areas within one tumor section up to a total of 8mm2 (total) (Fig 2). The microdissected sample was collected into the cap of a 0.2-mL tube.

Peptides were extracted using a Liquid Tissue™ MS Protein Prep kit (Expression Pathology Inc.), according to manufacturer's instructions. Briefly, each sample was suspended in 20 μL of Liquid Tissue™ buffer, incubated at 95°C for 90 min to digest protein crosslinks, cooled on ice for 2 min, and incubated with 1 μL trypsin-EDTA (0.25%, Thermo Fisher Scientific, Waltham, MA, USA) at 37°C overnight for peptide digestion. Finally, trypsin-digested samples were mixed with 2 μL of dithiothreitol and heated at 95°C for 5 min, and then stored at −20°C until analysis.

## LC-MS/MS-based shotgun proteomics

For LC-MS/MS analysis, we used a reversed-phase liquid chromatography (RP-LC) system (Ultimate 3000 nano LC) interfaced with an LTQ-Orbitrap hybrid mass spectrometer equipped with a nanoelectrospray ion source (all from Thermo Fisher Scientific). The RP-LC system comprised a C18 PepMap 100 trap column (length × inner diameter: 0.5 × 0.3 mm; Thermo Fisher Scientific) and a C18 tip column (length × inner diameter: 10 cm × 75 μm; particle diameter: 3 μm; Nikkyo Technos, Tokyo, Japan). Samples were loaded onto the trap column, washed with $H_2O$ containing 0.1% formic acid (solvent A) to concentrate and desalt them, and eluted using 95% acetonitrile, 5% $H_2O$ and 0.1% formic acid (solvent B). The 120-min LC gradient changed from 97.5% A/2.5% B to 77.5% A/22.5% B at 109 min, 65% A/35% B at 5 min, and finally 2% A/98% B at 2 min (at 0.3 μL/min). Eluted peptides were ionized by the electrospray and analyzed using the mass spectrometer (electrospray voltage: 1.8 kV; no sheath and auxiliary gas flow; capillary temperature: 250°C; collision energy: 35%; ion-selection threshold: 500 counts for MS/MS; Top N: 15; dynamic exclusion: 60 s).

## Protein identification and data analysis

For protein identification and quantification, raw data were searched using the MaxQuant software, version 1.6.0.16 (Max Planck Institute of Biochemistry, Planegg, Germany) and integrated using the Andromeda search engine [22–24]. MS spectra were analyzed by selecting N-terminal acetylation and methionine oxidation as variable modifications. Cysteine carbamidomethylation was chosen as the fixed modification. The reference database used was UniProt–SwissProt human canonical database version 2017–07 (canonical proteome). Minimum peptide length was set to seven amino acids, and 'match between runs' option was retained as default (match-time window: 0.7 min; alignment-time window: 20 min). Label-free quantification (LFQ) was enabled and LFQ minimum-ratio count was set to one. Remaining options were retained as default. Analysis was executed, and the 'ProteinGroups.txt' file was imported. The results were obtained from triplicate LC-MC/MC runs for each sample.

The imported file was analyzed using the Perseus software version 1.6.0.7 (Max Planck Institute of Biochemistry) [25]. Possible contaminants and reversed sequences were excluded. Protein-group LFQ intensities were $\log_2$-transformed. To overcome the error of missing LFQ values, missing values were imputed using random numbers drawn from a normal distribution in Perseus before fitting the models. Hierarchical clustering was performed on Z-score-normalized $\log_2$-transformed LFQ intensities. Log ratios were calculated as the difference in average $\log_2$ LFQ intensity values between $LN^-$ and $LN^+$ groups. The two groups were compared using the Student's $t$-test, and candidate proteins were found to be differentially expressed in a statistically significant manner ($\log_2$ ratio, $>1$ or $<1$; $p < 0.05$).

## Immunohistochemistry

For immunohistochemistry, 4-μm-thick FFPE tissue sections were deparaffinized in xylene and rehydrated in graded ethanol solutions and finally in distilled water. For antigen retrieval, sections were heated in 10 mM citrate/citric acid buffer (pH 6.0) for 15 min in a microwave oven or 5 min in an autoclave. Sections were incubated (overnight, 4˚C) with primary antibodies against AGR3 (1:2000, Abnova Co., Taipei, Taiwan), DEF3 (1:200, Abnova Co.), MYH14 (1:300, Novus Biologicals, Littleton, CO, USA), ABHD14B (1:250, OriGene Technologies GmbH, Herford, Germany), hemopexin (1:100, Gene Tex Inc., Irvine, CA, USA), FTL (1:1000, NSJ Bioreagents, San Diego, CA, USA), TPM1 (1:100, Abcam, Cambridge, UK), CSRP1 (1:100, Abcam), or plectin (1:250, Abcam). We searched the Protein Atlas website [26] for tissues documented to express these proteins strongly and used those as positive controls. After blocking endogenous peroxidase activity with 0.3% hydrogen peroxide in methanol, the labeled antigens were identified using the horseradish peroxidase EnVision+ System (DAKO, Glostrup, Denmark) and visualized using the chromogen 3,3′-diaminobenzidine tetrahydrochloride. Sections were lightly counterstained with hematoxylin.

All slides of immunostained sections were assessed by two of the authors (YS, TS) independently and in a blinded manner, and scored according to stroma intensity into four grades (0: negative, 1: weak, 2: moderate, 3: strong). In the case of assessor disagreement, the score was decided following discussion. For all statistical analyses, expression of each protein was dichotomized (0 and 1–3). The tissue location of protein expression was confirmed through comparison with sections of the same area stained with a marker for each tissue component: cytokeratin for cancer cells, vimentin and α-smooth muscle actin (αSMA) for CAF, leukocyte common antigen for lymphocytes, CD68 for macrophages, CD31 for vasculature, S100B protein for nerves, and αSMA for smooth muscle cells.

## Cell culture

For *in vitro* assays, we used two pancreatic-cancer cell lines, MIA PaCa-2 and Panc-1, which were cultured separately in RPMI 1640 medium (Sigma-Aldrich Co., Darmstadt, Germany) supplemented with 10% fetal bovine serum (FBS; Sigma-Aldrich Co.) and 1% penicillin/strep-tomycin (Thermo Fisher Scientific), at 37˚C in a humidified atmosphere containing 5% $CO_2$. The stimulant used was hemopexin purified from human plasma (Sigma-Aldrich Co.).

## Wound-healing assay

MIA PaCa-2 and Panc-1 cells were cultured until reaching confluence in 24-well plates in RPMI containing 10% FBS. After 14-h culture in serum-free medium (starvation), the cell layer in each well was carefully wounded using a P200 pipette tip, washed twice with PBS, and cultured in RPMI containing 0.5% FBS, alone or with 0.01–1 μM hemopexin, at 37˚C in 5% $CO_2$. Each wounded area was examined and measured using AxioVision version 4.8 (Cari Zeiss Co., Oberkochen, Germany) 12 and 24 h after wounding. Experiments were conducted in triplicate.

## Transwell® cell-migration and -invasion assays

Cell-migration and -invasion were assessed using the 24-well Transwell® plates featuring the control chamber and the Matrigel® invasion chamber (Corning, Corning, NY, USA). Cells were seeded (MIA PaCa-2 at $2 \times 10^5$/well; Panc-1 at $1 \times 10^4$/well) in 500 μL of serum-free RPMI into the upper chambers harboring 8-μm-pore membranes. Into the lower chambers, 750 μL of medium containing 10% FBS, without or with 0.01–1 μM hemopexin, was added. After a 22-h incubation at 37˚C in 5% $CO_2$, the upper-chamber media were discarded, cells adhering to the upper surface of the membrane were removed using a cotton applicator, and the cells that had passed through and adhered to lower surface of the membrane were fixed and stained using a Differential Quik Stain Kit (Modified Giemsa, Polysciences Inc., Warring-ton, PA, USA), according to the manufacturer's instructions. The membranes were cut out and covered until examination. Cells were counted in eight random fields (200×) selected for each membrane and the mean number of cells was calculated. Rate of invasion was %Invasion = (mean number of cells invading through Matrigel® insert membrane/mean number of cells migrating through control insert membrane) × 100. Experiments were performed in triplicate.

## Statistical analysis

For clinicopathological analyses, we used the JMP software version 14.0 (SAS Institute, Cary, NC, USA). In comparison between the two groups, significance was calculated using Pearson's $\chi^2$-test for categorical variates and Mann–Whitney $U$ test for continuous variates. $p < 0.05$ was considered statistically significant.

# Results

## Characteristics of patients whose specimens were selected for LC-MS/MS analysis

Table 1 lists the characteristics of patients whose specimens were selected for LC-MS/MS anal-ysis. No significant differences were present in age, sex, tumor location, tumor size, histological grade, R status, and adjuvant chemotherapy. Mean number of LN metastases in LN⁺ group was 9.2. In LN⁻ group, all patients had stage IIA cancer (UICC stage); in LN⁺ group, 2/5 had

**Table 1. Characteristics of patients whose samples were selected for LC-MS/MS analysis.**

| Characteristic | | LN⁻ group (n = 5) | LN⁺ group (n = 5) | p value |
|---|---|---|---|---|
| Age | Median | 61 (51–74) | 60 (54–74) | 0.7975 |
| Sex | M: F | 3:2 | 4:1 | 1.0000 |
| Location | Head | 3 | 2 | |
| | Body | 2 | 1 | 0.4603 |
| | Whole | 0 | 2 | |
| Size (mm) | Mean±SD | 28±4.5 | 37±9.4 | 0.0894 |
| Number of LN metastases | Mean±SD | 0 | 9.2±1.2 | **0.0001** |
| UICC Stage | IIA | 5 | 0 | |
| | III | 0 | 2 | **0.0079** |
| | IV | 0 | 3 | |
| Histological grade | G1 | 1 | 1 | 1.0000 |
| | G2 | 4 | 4 | |
| OS (months) | Median | 66 (44.4–146) | 13 (6.5–13.5) | **0.0089** |
| R status | R0 | 5 | 5 | |
| Adjuvant chemotherapy | GEM | 4 | 4 | 1.0000 |
| | S1 | 1 | 0 | |
| | None | 0 | 1 | |

Staging was based on UICC 8th edition.

GEM, gemcitabine; LC-MS/MS, liquid chromatography–tandem mass spectrometry; LN, lymph node; OS, overall survival.

stage III, and 3/5 had stage IV due to para-aortic LN metastasis. The median overall survival (OS) was 66 months in the LN⁻ group and 13 months in the LN⁺ group ($p$ = 0.0089).

## LC-MS/MS-based protein identification in PDAC stroma

LC-MS/MS-based shotgun proteomic analysis using the PDAC stroma resulted in the detection of 445 and 396 proteins in LN⁻ and LN⁺ groups, respectively. In total, 490 proteins were identified, 351 being detected in both groups (Fig 3A). The proteomic data were analyzed using the Perseus software for group comparison, and the volcano plot of identified proteins was imported (Fig 3B). Candidate proteins were selected using these criteria: (1) differential expression between the groups, based on $\log_2$ ratio of $>1$ or $<1$, and (2) statistically significant difference ($p < 0.05$, Student's $t$-test) (shaded area, Fig 3B). Nine proteins were differentially expressed between the groups (red plot, Fig 3B). Four proteins including anterior gradient 3, protein disulfide isomerase family member (AGR3); α-defensin 3 (DEF3); myosin heavy chain 14 (MYH14); and abhydrolase domain containing 14B (ABHD14B) were overexpressed in the LN⁻ group. Five proteins, hemopexin, ferritin light chain (FTL), α-tropomyosin (TPM1), cysteine- and glycine-rich protein 1 (CSRP1), and plectin were overexpressed in the LN⁺ group (S1 Table).

## Immunohistochemical validation of protein expression

We performed immunohistochemistry on PDAC tissue sections to confirm expression of the nine candidate proteins identified through LC-MS/MS analysis of specimens from the 10 patients (preceding subsection). Antibodies against each protein successfully stained the positive controls. Among the nine proteins, TPM1 and MYH14 were not detected in any sample, but the other proteins were expressed at varying levels in the stromal tissue. Notably, immunohistochemistry of hemopexin and FTL concurred well with LC-MS/MS results: stromal

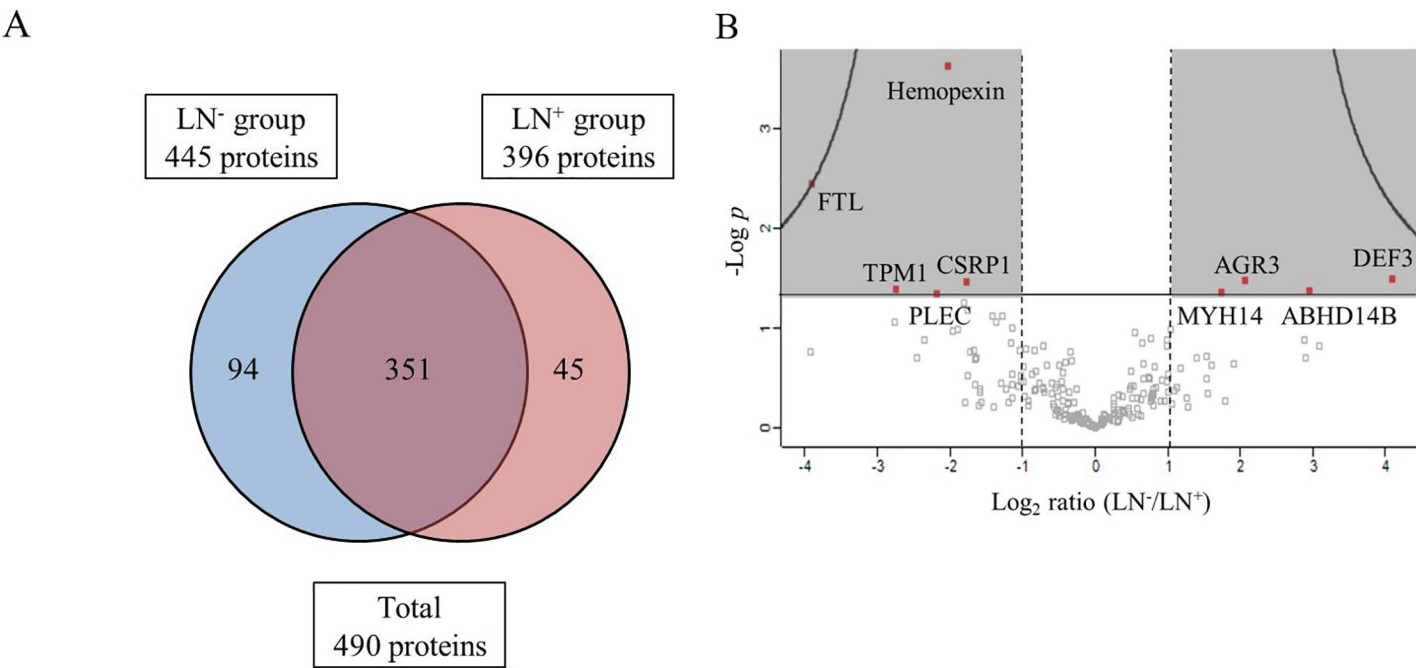

**Fig 3. Proteomics analysis by LC-MS/MS using the PDAC stroma.** (A) Venn diagram of proteins detected in LN⁻ and LN⁺ groups by using LC-MS/MS analysis; 490 proteins were detected in total, 351 of which were present in both groups. (B) Volcano plot of detected proteins (imported from Perseus software). Vertical axis: level of significance; horizontal axis: logarithm of the ratio of the amounts of each protein detected in LN⁻ or LN⁺ groups. Red plots in shaded area, satisfying $\log_2$ ratio of $>1$ or $<1$ and also showing statistical significance ($p < 0.05$, Student's $t$-test), are proteins exhibiting significantly different abundances between the two groups. In total, nine proteins were selected as candidate proteins.

expression of both proteins was higher in LN⁺ group than LN⁻ group according to LC-MS/MS results, and both proteins showed robust expression also by immunohistochemical validation (Fig 4A). Thus, hemopexin and FTL were selected as final candidate proteins for further study. Hemopexin and FTL were mainly expressed in the cancer-cell cytoplasm and stromal fibroblasts (Fig 4B). They showed focal expression in lymphocytes and macrophages, but no expression in vascular endothelium.

Hemopexin and FTL were more strongly expressed at stroma in the LN⁺ group than LN⁻ group (A). Hemopexin was expressed in the cancer-cell cytoplasm and in stromal fibroblasts (B), FTL expression was localized in the same location as hemopexin (not shown here). Magnification: 100×. Scale bar: 100 μm.

## Characteristics of patients selected for clinicopathological analysis

Among the 163 patients, 103 and 60 were male and female, and 35 and 128 were LN⁻ and LN⁺, respectively. Other recorded characteristics included mean tumor size, 31 mm; mean LN-metastasis number, 2.5 in all; UICC staging, IA/IB:IIA:IIB:III:IV = 28 (17%):7 (4%):77 (47%):22 (14%):29 (18%); R status, R0:R1:R2 = 145 (89%):15 (9%):3 (2%); median OS, 23 months.

## Relationship between hemopexin expression and clinicopathological parameters

Hemopexin and FTL expression was analyzed using immunohistochemistry, and the correlation between protein expression level and clinicopathological parameters was investigated.

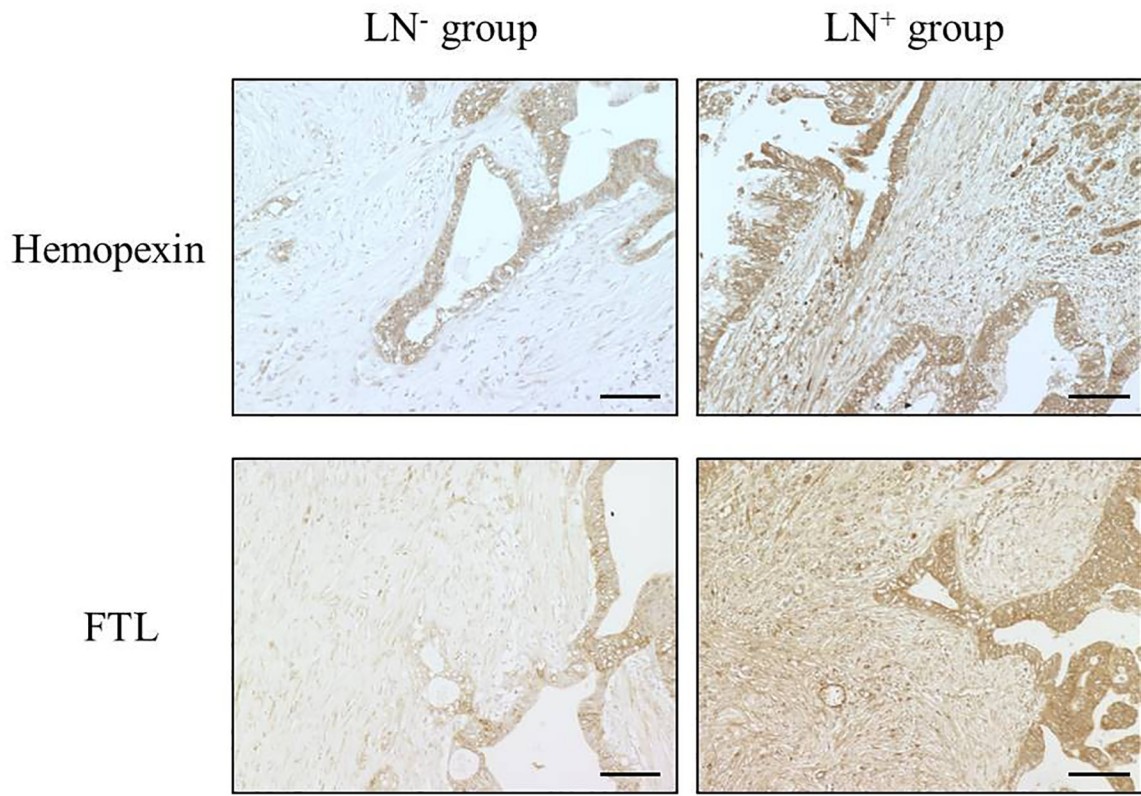

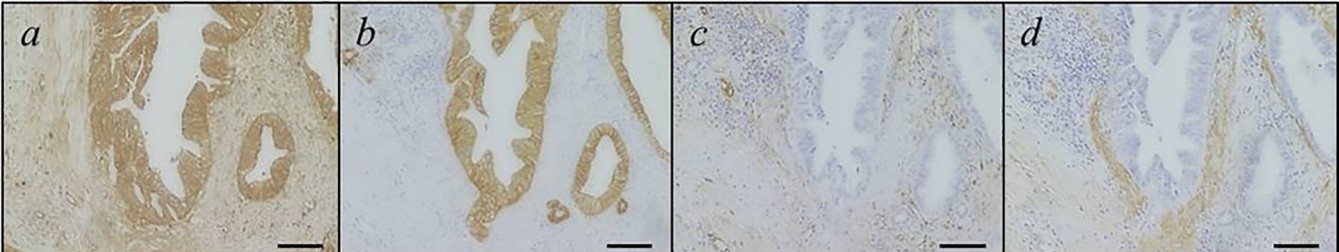

**Fig 4. Immunohistochemistry of hemopexin and ferritin light chain (FTL).** (A) Representative images of immunohistochemistry of hemopexin and FTL in LN⁻ and LN⁺ groups. Magnification: 100×. Scale bar: 100 μm. (B) Representative images of marker staining for localization diagnosis by using serial sections of LN⁺ group samples. *a*: Immunostained section of hemopexin-positive sample. *b–d*: Sections immunostained for marker of each component (*b*: cytokeratin; *c*: vimentin; *d*: αSMA).

Hemopexin expression was associated with several clinicopathological parameters associated with LN metastasis, whereas FTL expression was not associated with those parameters. Thus, we performed univariate analysis on hemopexin expression and each parameter.

Table 2 shows the results of clinicopathological analysis of hemopexin expression. Among the 163 PDAC patients, 26 (16%) and 137 (84%) were hemopexin-negative and hemopexin-

**Table 2. Clinicopathological analysis of hemopexin expression in 163 patients diagnosed with invasive PDAC who had received no neoadjuvant therapy.**

| Variable | | Hemopexin⁻ group (n = 26) | Hemopexin⁺ group (n = 137) | *p* value |
|---|---|---|---|---|
| Age | Median | 66 (27–82) | 68 (33–88) | 0.1366 |
| Sex | Male | 16 (62%) | 87 (64%) | 0.8489 |
| | Female | 10 (38%) | 50 (36%) | |
| Location | Head | 23 (88%) | 80 (59%) | **0.0132** |
| | Body | 2 (8%) | 47 (34%) | |
| | Whole | 1 (4%) | 10 (7%) | |
| Size (mm) | Mean±SD | 31±10.3 | 31±11.6 | 0.7925 |
| R status | R0 | 23 (88%) | 122 (89%) | 0.0796 |
| | R1 | 1 (4%) | 14 (10%) | |
| | R2 | 2 (8%) | 1 (1%) | |
| T stage | ≤2 | 19 (73%) | 104 (76%) | 0.7581 |
| | ≥3 | 7 (27%) | 33 (24%) | |
| N stage | ≤1 | 24 (92%) | 101 (74%) | **0.0399** |
| | 2 | 2 (8%) | 36 (26%) | |
| M stage | 0 | 22 (85%) | 112 (82%) | 0.7263 |
| | 1 | 4 (15%) | 25 (18%) | |
| LN ratio | Mean±SD | 0.05±0.04 | 0.12±0.15 | **0.0252** |
| Histological grade | G1 | 4 (15%) | 25 (18%) | 0.6250 |
| | G2 | 21 (81%) | 100 (73%) | |
| | G3 | 1 (4%) | 12 (9%) | |
| Venous invasion | ≤2 | 20 (77%) | 67 (50%) | **0.0096** |
| | 3 | 6 (23%) | 69 (50%) | |
| Lymphatic invasion | ≤2 | 25 (96%) | 105 (77%) | **0.0232** |
| | 3 | 1 (4%) | 32 (23%) | |
| Nerve invasion | ≤2 | 16 (62%) | 69 (50%) | 0.2957 |
| | 3 | 10 (38%) | 68 (50%) | |

TNM classification was based on UICC 8th edition. Tumor-invasion grade was based on the Classification of Pancreatic Carcinoma, 4th English edition, by Japan Pancreas Society. The degrees of venous invasion (v), lymphatic invasion (ly), and nerve invasion (ne) were classified into four grades (0: negative; 1: minor; 2: moderate; 3: severe) according to pathologists' assessment.

LN, lymph node; PDAC, pancreatic ductal adenocarcinoma.

positive, respectively. Hemopexin staining pattern did not differ with age, sex, tumor size, UICC T and M stages, and histological grade, but tumor location differed significantly between the two groups. Analysis of LN-metastasis-associated parameters revealed that in the hemopexin-positive group, the proportion of UICC N2 (i.e., LN-metastasis number $\geq$ 4) was significantly higher than that of UICC N0+N1 (26.3% vs. 7.7%, $p = 0.0399$), and the LN ratio (number of positive LN/number of dissected LN) was also higher (0.12 vs. 0.05, $p = 0.0252$). Expression scores of hemopexin in cases with lymph node metastasis were 0 in 21, 1 in 37, 2 in 54 and 3 in 16 cases (N1; 0 in 18, 1 in 28, 2 in 36 and 3 in 8 cases, N2; 0 in 2, 1 in 10, 2 in 18 and 3 in 8 cases). Furthermore, we comprehensively analyzed the tumor-invasion grade as per the Classification of Pancreatic Carcinoma [8] by the Japan Pancreas Society, according to which the degrees of venous invasion (v), lymphatic invasion (ly), and nerve invasion (ne) were classified into four grades each (0: negative; 1: minor; 2: moderate; 3: severe) based on the assessment of pathologists. Here, venous-invasion and lymphatic-invasion grades were significantly higher in the hemopexin-positive group than in the hemopexin-negative group (v3: 50.1% vs. 23.1%, $p = 0.0096$; ly3: 23.4% vs. 3.9%, $p = 0.0232$). Nerve invasion was also generally

more severe in the hemopexin-positive group, but the difference was not statistically significant. These results suggested that hemopexin expression in tumor tissue was potentially associated with LN metastasis and invasion of PDAC.

### Effect of hemopexin on wound healing, cell-migration and cell-invasion

We performed wound-healing assays and Transwell®-based cell-migration and cell-invasion assays to assess how hemopexin affects cell migration and invasion.

The wound-healing assay (Fig 5A and 5B) revealed increased cell migration into the wound area with increasing hemopexin concentration 24 h after wounding. In both cell lines (MIA PaCa-2 and Panc-1), the wound-repair rate (%Repair) increased as the hemopexin concentration was increased (Fig 5B). At 1 μM, hemopexin significantly promoted migration of the MIA PaCa-2 cells relative to control at 12 and 24 h ($p$ = 0.0043 and 0.0287, respectively), and this was also observed at 0.1 μM hemopexin ($p$ = 0.0104 and 0.0458, respectively), but to a lesser extent than that at 1 μM. At the highest concentration of hemopexin, a 20% increase in Panc-1 migration was observed, although this did not reach statistical significance.

The cell-migration and -invasion assays yielded these results (Fig 5C and 5D): with both cell lines, the number of invading cells, observed microscopically, increased markedly with increasing hemopexin concentration (Fig 5C), whereas the number of migrating cells increased slightly with increasing hemopexin concentration (upper graphs, Fig 5D). Thus, the ratio of invading cells (%Invasion) at 0.1 and 1 μM hemopexin was significantly higher than that in control in both cell lines (lower graphs, Fig 5D). Hemopexin significantly increased % Invasion of MIA PaCa-2 cells relative to control at 0.1 and 1 μM ($p$ = 0.0129 and 0.0036, respectively), and this was also observed in Panc-1 cells ($p$ = 0.0133 and 0.0032, respectively). These results demonstrated that hemopexin enhanced the invasive ability of the pancreatic-cancer cells and also tended to promote cell migration.

### Discussion

Stromal hyperplasia is a cardinal histological feature of PDAC–stroma surrounds 30%–90% of the tumor cells [18]. The tumor stroma promotes cancer progression by interacting with epithelial cells and inhibits response to chemotherapy and radiation by acting as a physical barrier to cancer cells [27]. Accordingly, diverse stroma-targeting therapies have been developed, some of which are now in clinical trials [28–30]. However, the comprehensive protein expression in PDAC stroma remains poorly elucidated.

Recently, the tumor microenvironment has attracted attention as a proteomics target in cancer research, and the PDAC microenvironment has been investigated in a few studies. The secreted proteome was analyzed using a pancreatic stellate cell line or body fluids such as pancreatic juice and plasma [31,32], and in three studies, the proteome of pancreatic-cancer stromal tissue, collected using LMD, was analyzed and compared to those of cancer cells [33–35]. However, to our knowledge, the PDAC stromal tissue proteome had not been studied. Previous studies examined protein-expression differences between cancer cells and stroma to elucidate cancer–stroma interactions. However, in certain PDAC patients, no LN metastasis is detected, and the disease progresses slowly, whereas in others, several metastases are present, and disease progression is rapid. To ascertain what causes clinical aggressiveness, comparing patient groups with distinct clinical characteristics or prognoses is necessary. Therefore, we compared the PDAC stromal proteome from LN⁻ and LN⁺ patients to test the hypothesis that protein expression might differ between patients with divergent LN-metastasis status. Thus, in this study, we demonstrate, for the first time, by proteomic analysis of the tumor stroma, that hemopexin is a previously unrecognized LN-metastasis-associated protein of PDAC.

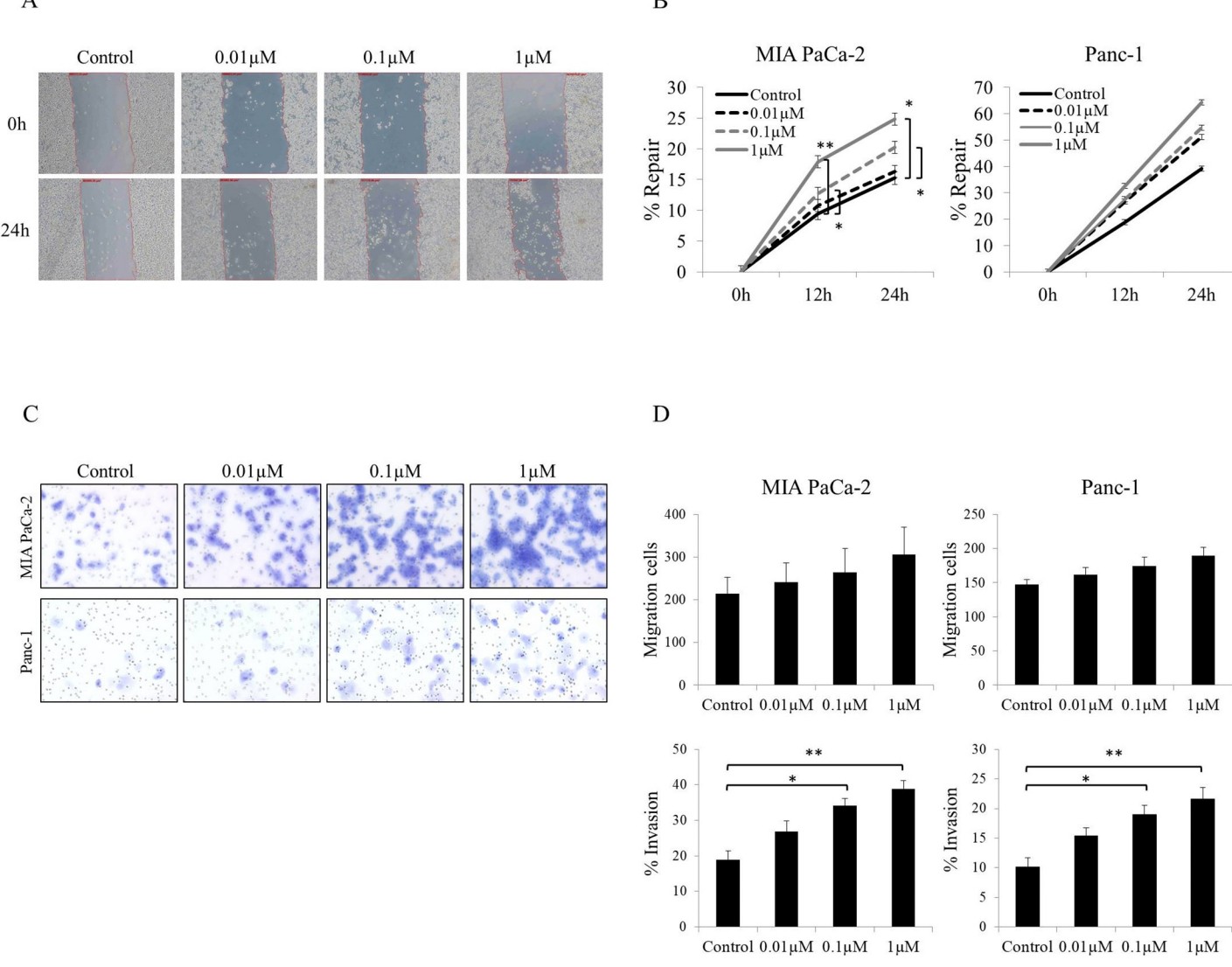

**Fig 5. Wound-healing, cell-migration, and cell-invasion assays.** (A) Representative micrographs of MIA PaCa-2 cells at 0 or 24 h after wounding. Cells were cultured in 0.5% FBS/RPMI alone or medium including 0.01–1 μM hemopexin. Magnification: 50×. (B) Wound-repair rate (%Repair) of MIA PaCa-2 and Panc-1 cells; in both cell lines, %Repair increased with an increase in hemopexin concentration at 12 or 24 h. In MIA PaCa-2 cells, %Repair at 0.1 or 1 μM hemopexin was significantly increased relative to control at 12 or 24 h. (C, D) Migration and invasion assays performed using Transwell®. (C) Representative micrographs of invading cells. Upper images: MIA PaCa-2 cells; lower panels: Panc-1 cells. With both cell lines, invading cells increased with an increase in hemopexin concentration. Magnification: 200×. (D) Number of migrating cells and percentage of invading cells (%Invasion). In both cell lines, the number of migrating cells increased slightly as the hemopexin concentration was increased, but the change was not statistically significant. %Invasion also increased with an increase in hemopexin concentration; 0.1 and 1 μM hemopexin significantly promoted cell invasion as compared to control treatment in both cell lines. %Invasion = (mean number of cells invading through Matrigel® insert membrane/mean number of cells migrating through control insert membrane) × 100. Cells were counted in eight random fields (200×) and the mean number was calculated. Each experiment was conducted in triplicate. $^{**}p < 0.01$, $^{*}p < 0.05$.

We used a two-step process for selecting candidate proteins: we first performed software analysis to select nine proteins expressed differentially between LN⁻ and LN⁺ groups, and then from these nine candidates, we selected the final candidates through immunohistochemistry in the 10 patients whose specimens were subjected to LC-MS/MS analysis. In the volcano plot generated using software analysis, hemopexin showed the highest significance level among the nine proteins. Conversely, FTL featured the largest $\log_2$ ratio among candidate proteins over-expressed in LN⁺ group, as well as the second-highest significance level (after hemopexin).

Thus, these two notable proteins were selected as final candidates because they showed concordant immunohistochemical results. Some proteins identified by proteomics were not stained by immunohistochemistry. This might be due to effects of antibody quality or protein amount. Since candidate proteins were selected by relative comparison between the two groups, the amount of protein might not be enough for immunohistochemistry.

Clinicopathological analysis revealed that hemopexin expression was associated with multiple LN-metastasis-associated parameters. However, FTL showed no association with any parameter in the large cohort despite the compatible validation results obtained with the smaller discovery groups (preceding paragraph). Two reasons might account for this; the first is the small sample size of discovery groups. Because LFQ intensity is calculated as the sum within the group, if the sample size is small, deviations in spectral-detection levels between samples might represent large differences in appearance. The second reason is the smaller range of detectable difference between immunohistochemistry and MS. In LC-MS/MS analysis, even a very weakly detected protein might be selected as a differentially expressed protein if the difference is considered adequately large in the group comparison. Expression difference was clearly observed in the discovery groups because the difference in LN-metastasis numbers was sufficiently large, but slight differences in immunohistochemical detection might not be readily reflected in a large cohort. Therefore, differentially expressed candidate proteins identified in a small cohort do not unfailingly show the same result in a large cohort. Immunohistochemical confirmation of expression in practice is critical for assessing whether a protein is a valid candidate.

Hemopexin overexpression was found here to be associated with LN-metastasis number, venous invasion, lymphatic invasion, and LN ratio, which is also an accurate indicator of prognosis [36,37]. Intriguingly, tumor size did not differ between the groups, suggesting that hemopexin is more closely associated with invasion and metastasis than to tumor proliferation. This view was bolstered by the results of *in vitro* assays showing that hemopexin strongly promoted the invasive ability of the pancreatic-cancer cells.

Hemopexin is a glycoprotein produced mainly by the liver and secreted into plasma, which contains hemopexin at high concentrations (0.5–1 mg/mL). Hemopexin exhibits high hemoglobin-binding affinity, functions as a scavenger of free heme in plasma (which is harmful to the human body) and prevents free-radical reactions [38]. Hemopexin is reportedly expressed in neurons, cerebrospinal fluid, and retina [39–41]. Besides heme-scavenging, other functions of hemopexin include iron homeostasis, antioxidant protection, anti-inflammatory effects, signaling to promote cell survival, and gene expression [42,43]. Recently, hemopexin was demonstrated to promote neural stem-cell migration and differentiation into neurons and oligodendrocytes *in vitro* [44]. In relation to cancer, hemopexin was reported to be upregulated in pleural effusions of lung-cancer patients and in plasma of pancreatic-cancer patients before resection [45,46], and the fucosylated form of hemopexin was reported as an accurate serum marker for hepatocellular carcinoma [47,48]. However, hemopexin expression in cancer tissue has not been previously reported, and how hemopexin functions in cancer remains unclear.

An important question to ask is how hemopexin promotes invasion of PDAC. Low-density lipoprotein receptor-related protein-1 (LRP1) reportedly acts as a receptor for hemopexin [43], and its expression on pancreatic cancer cells was confirmed [49]. LRP1 is a receptor protein that binds to the heme–hemopexin complex with high affinity, but apo-hemopexin can also bind LRP1. We postulated that hemopexin present in the stroma of PDAC binds to LRP1 on the cancer-cell membrane, thereby promoting invasion by activating the mitogen-activated protein-kinase pathway and the NF-κβ pathway [50]. In addition, overexpression of hemopexin activates the tumor-associated matrix metalloproteinases (MMPs) [51], enzymes that

promote tumor progression by facilitating cancer invasion and metastasis. Intriguingly, MMPs contain hemopexin-like domains [52]. However, whether MMP and hemopexin share common physiological roles is still unknown. Although expressions of MMPs or LRP1 were not evaluated in the current study, they may affect the function of hemopexin.

Limitations of our study are the following. First, hemopexin function was not adequately investigated: how hemopexin mechanistically contributes to tumor progression is unclear. Hemopexin function should be confirmed and reproduced *in vivo*. Second, hemopexin-secretion sites were not identified. We performed *in vitro* assays based on the hypothesis that hemopexin is secreted from CAF and affects cancer cells. However, immunohistochemistry revealed strong hemopexin expression in not only fibroblasts but also cancer-cell cytoplasm. Thus, hemopexin production potentially by cancer cells should be considered, and whether hemopexin affects CAF or other tumor-microenvironment components should be investigated. Hemopexin may have paracrine and autocrine mechanisms with crosstalk of cancer cells and CAF. Third, the association between hemopexin overexpression and PDAC prognosis is unknown. The results suggested that high hemopexin expression is associated with poor prognosis. Here, we selected patients who had not received neoadjuvant therapy so as to avoid histological outcomes, such as necrosis and degeneration. However, the proportion of patients receiving neoadjuvant therapy has been increasing recently because of clinical trials at our facility. For prognostic analysis, selecting a larger cohort, including patients with neoadjuvant therapy, is desirable.

The results of our proteomic analyses on FFPE tissues and *in vitro* assays have suggested for the first time that hemopexin activates cell invasion and metastasis in PDAC. These results provide new insights into hemopexin function in pancreatic-cancer progression. Hemopexin could serve as a novel therapeutic target or biomarker for PDAC, in which prognosis is poor when the LN-metastasis number is large. For practical applications of hemopexin, further investigation must be conducted to elucidate the mechanisms by which hemopexin activates tumor invasion.

## Supporting information

**S1 Table. Fold change and significance for the candidate proteins.** Fold change and significance for 9 proteins selected on the volcano plot. AGR3, anterior gradient 3, protein disulfide isomerase family member; DEF3, α-defensin 3; MYH14, myosin heavy chain 14; ABHD14B, abhydrolase domain containing 14B; FTL, ferritin light chain; TPM1, α-tropomyosin; CSRP1, cysteine- and glycine-rich protein 1.
(DOCX)

## Acknowledgments

We thank Emiko Shibuya and Keiko Inabe (Department of Surgery, Tohoku University) for technical assistance.

## Author Contributions

**Conceptualization:** Yukie Suzuki, Tatsuyuki Takadate.

**Data curation:** Yukie Suzuki, Hiroki Shima, Takashi Suzuki.

**Formal analysis:** Yukie Suzuki, Hiroki Shima.

**Funding acquisition:** Tatsuyuki Takadate.

**Methodology:** Yukie Suzuki, Hiroki Shima, Takashi Suzuki, Mitsuhiro Shimura.

**Project administration:** Yukie Suzuki.

**Software:** Hiroki Shima.

**Supervision:** Tatsuyuki Takadate, Masamichi Mizuma, Hiroki Shima, Tomoyoshi Tachibana, Mitsuhiro Shimura.

**Validation:** Yukie Suzuki, Takashi Suzuki, Tomoyoshi Tachibana.

**Visualization:** Yukie Suzuki.

**Writing – original draft:** Yukie Suzuki.

**Writing – review & editing:** Yukie Suzuki, Tatsuyuki Takadate, Masamichi Mizuma, Tatsuo Hata, Masahiro Iseki, Kei Kawaguchi, Takeshi Aoki, Hiroki Hayashi, Takanori Morikawa, Kei Nakagawa, Fuyuhiko Motoi, Takeshi Naitoh, Kazuhiko Igarashi, Michiaki Unno.

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
