## [Decision Letter · Decision Letter 0]

27 Apr 2020

PONE-D-20-00932

Stromal expression of hemopexin is associated with lymph-node metastasis in pancreatic ductal adenocarcinoma

PLOS ONE

Dear ズキユキエ ス,

Thank you for submitting your manuscript to PLOS ONE. After careful consideration, we feel that it has merit but does not fully meet PLOS ONE’s publication criteria as it currently stands. Therefore, we invite you to submit a revised version of the manuscript that addresses the points raised during the review process.

We would appreciate receiving your revised manuscript by 8th June 2020. To enhance the reproducibility of your results, we recommend that if applicable you deposit your laboratory protocols in protocols.io, where a protocol can be assigned its own identifier (DOI) such that it can be cited independently in the future. For instructions see: http://journals.plos.org/plosone/s/submission-guidelines#loc-laboratory-protocols

We look forward to receiving your revised manuscript.

Kind regards,

Eithne Costello

Academic Editor

PLOS ONE

Journal Requirements:

2. In the ethics statement in the manuscript and in the online submission form, please provide additional information about the patient records used in your retrospective study, including: a) whether all data were fully anonymized before you accessed them and b) the date range (month and year) during which patients' medical records were accessed."

3. At this time, we ask that you please provide the product numbers and lot numbers of the primary antibodies used for the immunohistochemical analysis in this study

In addition, please provide the product number and lot number of the hemopexin purchased from Sigma Aldrich in your study."

4. Please provide additional information about each of the cell lines used in this work, including source, history  and any quality control testing procedures (authentication, characterisation, and mycoplasma testing). For more information, please see " ext-link-type="uri" xlink:type="simple">http://journals.plos.org/plosone/s/submission-guidelines#loc-cell-lines."

5. To comply with PLOS ONE submission guidelines, in your Methods section, please provide additional information regarding your statistical analyses. For more information on PLOS ONE's expectations for statistical reporting, please see https://journals.plos.org/plosone/s/submission-guidelines.#loc-statistical-reporting

6. Thank you for stating in your Funding Statement:

"This work was supported, in part, JSPS(Japan Society For The Promotion Of Science

URLhttps://www.jsps.go.jp/j-grantsinaid/16_rule/rule_h24.html) KAKENHI Grant

number 19K18107. Tatsuyuki Takadate received the grant.

The funder had no role in study design, data collection and analysis, decision to

publish, or preparation of the manuscript."

7. PLOS requires an ORCID iD for the corresponding author in Editorial Manager on papers submitted after December 6th, 2016. Please ensure that you have an ORCID iD and that it is validated in Editorial Manager. To do this, go to ‘Update my Information’ (in the upper left-hand corner of the main menu), and click on the Fetch/Validate link next to the ORCID field. This will take you to the ORCID site and allow you to create a new iD or authenticate a pre-existing iD in Editorial Manager. Please see the following video for instructions on linking an ORCID iD to your Editorial Manager account: https://www.youtube.com/watch?v=_xcclfuvtxQ

Reviewers' comments:

Reviewer's Responses to Questions

**Comments to the Author**

1. Is the manuscript technically sound, and do the data support the conclusions?

Reviewer #1: Yes

Reviewer #2: Partly

2. Has the statistical analysis been performed appropriately and rigorously? 

Reviewer #1: Yes

Reviewer #2: I Don't Know

3. Have the authors made all data underlying the findings in their manuscript fully available?

Reviewer #1: Yes

Reviewer #2: Yes

4. Is the manuscript presented in an intelligible fashion and written in standard English?

Reviewer #1: Yes

Reviewer #2: No

5. Review Comments to the Author

Reviewer #1: The manuscript submitted by Suzuki et al performed the proteomic analysis from the stromal compartment of the pancreatic cancer tissues derived from the patients with and without LN metastasis. The authors subsequently focused on the hemopexin and demonstrated that patients with LN metastasis express higher levels of hemopexin in their tumor tissues. The expression of hemopexin also significantly correlated with location, N-stage, LN ratio, venous and lymphatic invasion. The experiments performed in vitro with two pancreatic cancer cell lines, MIA PaCa-2 and Panc1 showed that hemopexin increased the invasive potential of the pancreatic cancer cells. The manuscript is well written and the findings support the overall conclusions. However, I have the following concerns.

Major concerns

1. Authors suggested that cancer-associated fibroblast (CAF) may be the predominant source of the hemopexin; however, immunohistochemistry analysis in Fig. 4A suggest that hemopexin may be predominantly contributed by the pancreatic cancer cell, hence, the overall mechanism may by autocrine instead of paracrine (cancer CAF crosstalk).

2. Why Panc-1 cells showed a difference only in the invasion but not in migration? What is the status of LRP1 (CD91) expression on MIA PaCa-2 and Panc-1 cells? What about the difference in MMPs expression between two cell lines after treatment with hemopexin?

3. How many serial samples were stained (IHC) for the clinicopathological analysis is not clearly mentioned. What was the range of hemopexin expression in the LN metastatic samples?

4. No need to show two volcano plots in Fig. 3B.

5. Include a table of fold change and significance in the supplementary materials.

Reviewer #2: This manuscript presents data from proteomic profiling of stromal region proteins that were collected from pancreatic tumor sections of patients with or without lymph node metastasis. Out of a total of 490 proteins, 9 were differentially expressed between the two patient groups. Further, authors performed immunohistochemistry (IHC) of one protein, hemopexin and assessed its clinicopathological correlations. Some interesting correlations were observed and authors also found in in vitro studies that hemopexin promoted invasion of the pancreatic cancer cells. Overall, findings are novel and interesting, but manuscript writing is not up to the mark and needs significant improvement. Additionally, following concerns should be addressed:

1. Following sentences in the abstract need to be rearranged and refined – “Certain factors……..profiling of tumor stroma.” should be changed to “Certain protein…………...invasion and metastasis.”, and moved after first sentence. “We selected….. and 2015” should be changed to “we selected……and 2015 and subjected to comprehensive proteomic profiling of tumor stroma”.

2. It would be best to collect stromal areas from different areas within one tumor section since there could be significant stromal heterogeneity.

3. How many technical replicates were taken in the proteomic analyses?

4. In IHC analyses, discuss why the identified stromal proteins show minimal stromal staining?

5. All figure legends should have been written separately.

6. PLOS authors have the option to publish the peer review history of their article (what does this mean?). If published, this will include your full peer review and any attached files.

Reviewer #1: No

Reviewer #2: No

---

## [Author Response · Author response to Decision Letter 0]

10 Jun 2020

PONE-D-20-00932

Stromal expression of hemopexin is associated with lymph-node metastasis in pancreatic ductal adenocarcinoma

Authors: Suzuki et al.

Dear Joerg Heber, Editor-in-Chief of PLOS ONE:

We appreciate the time and all of the suggestions from the reviewers. Their comments have helped us to improve the manuscript. We have attempted to address the comments as completely as possible, as outlined below:

To Editor:

Response:

Yes, I confirmed.

2. In the ethics statement in the manuscript and in the online submission form, please provide additional information about the patient records used in your retrospective study, including: a) whether all data were fully anonymized before you accessed them and b) the date range (month and year) during which patients' medical records were accessed."

Response:

We added the sentence in the metods section as follows. “All data were fully anonymized before we accessed them. Patients' medical records were accessed between April 2016 to September 2018.”

3. At this time, we ask that you please provide the product numbers and lot numbers of the primary antibodies used for the immunohistochemical analysis in this study

In addition, please provide the product number and lot number of the hemopexin purchased from Sigma Aldrich in your study."

Response:

I've attached the product numbers and lot numbers of the primary antibodies and hemopexin. But the lot number of FRIL antibody is not recorded.

4. Please provide additional information about each of the cell lines used in this work, including source, history and any quality control testing procedures (authentication, characterisation, and mycoplasma testing). 

Response:

I've attached the cell line authentication.

5. To comply with PLOS ONE submission guidelines, in your Methods section, please provide additional information regarding your statistical analyses. 

Response: 

Yes, I confirmed.

6. Thank you for stating in your Funding Statement:

"This work was supported, in part, JSPS(Japan Society For The Promotion Of Science URLhttps://www.jsps.go.jp/j-grantsinaid/16_rule/rule_h24.html) KAKENHI Grant number 19K18107. Tatsuyuki Takadate received the grant. The funder had no role in study design, data collection and analysis, decision to publish, or preparation of the manuscript."

Please provide an amended statement that declares *all* the funding or sources of support (whether external or internal to your organization) received during this study. Please also include the statement “There was no additional external funding received for this study.” in your updated Funding Statement. Please include your amended Funding Statement within your cover letter. We will change the online submission form on your behalf.

Response:

We added the sentence in the acknowledgments section as follows. “This work was supported, in part, JSPS (Japan Society For The Promotion Of Science URLhttps://www.jsps.go.jp/j-grantsinaid/16_rule/rule_h24.html) KAKENHI Grant number 16K19914 and 19K18107. Tatsuyuki Takadate received the grant. The funder had no role in study design, data collection and analysis, decision to publish, or preparation of the manuscript. There was no additional external funding received for this study.”

7. PLOS requires an ORCID iD for the corresponding author in Editorial Manager on papers submitted after December 6th, 2016. Please ensure that you have an ORCID iD and that it is validated in Editorial Manager. 

Response:

Yes, I confirmed.

To Reviewer #1: 

Major concerns

1. Authors suggested that cancer-associated fibroblast (CAF) may be the predominant source of the hemopexin; however, immunohistochemistry analysis in Fig. 4A suggest that hemopexin may be predominantly contributed by the pancreatic cancer cell, hence, the overall mechanism may by autocrine instead of paracrine (cancer CAF crosstalk).

Response:

Thank you for the comment of critical importance. We agree on the reviewer’s comment. The overall mechanism of hemopexin secretion may be crosstalk of cancer cells and CAF. We added the sentences in the discussion section, as follows. “Hemopexin may have paracrine and autocrine mechanisms with crosstalk of cancer cells and CAF.”

2. Why Panc-1 cells showed a difference only in the invasion but not in migration? What is the status of LRP1 (CD91) expression on MIA PaCa-2 and Panc-1 cells? What about the difference in MMPs expression between two cell lines after treatment with hemopexin?

Response:

Thank you for the important comment. Migration of the Panc-1 cells was not significantly different between control cells and cells exposed with each concentration of hemopexin. However, migration of ability in cells exposed with 1 μM of hemopexin was 20% higher than that in control. Thus, we think there seems to be a similar tendency to MIA PaCa-2 in the migration. We rewrote the sentences of the result section as follows. “Panc-1 migration did not significantly differ from control at each hemopexin concentration. However, it was 20% higher than control at 1μM hemopexin. There seemed to be a similar tendency to MIA PaCa-2 in the migration.” 

MMPs and LRP1 were only considered in the literature, and their expression was not investigated in this study. We added the sentence in the discussion section as follows. “Although expressions of MMPs or LRP1 were not evaluated in the current study, they may affect the function of hemopexin.” 

3. How many serial samples were stained (IHC) for the clinicopathological analysis is not clearly mentioned. What was the range of hemopexin expression in the LN metastatic samples?

Response:

We are sorry for not clearly describing number of serial samples stained for the clinicopathological analysis. It had been described in “Characteristics of patients selected for clinicopathological analysis” of the results section in the previous manuscript, as follows. “For clinicopathological analysis, we selected 163/183 patients with histologically diagnosed invasive PDAC; the patients had received no neoadjuvant therapy. We excluded the 10 patients whose specimens were selected for LC-MS/MS analysis and 10 whose specimens were unavailable.” We moved these sentences to the method section in the revised manuscript. 

Expression scores of hemopexin in cases with lymph node metastasis were 0 in 21, 1 in 37, 2 in 54 and 3 in 16 cases (N1; 0 in 18, 1 in 28, 2 in 36 and 3 in 8 cases, N2; 0 in 2, 1 in 10, 2 in 18 and 3 in 8 cases). This sentence was added in the result section. 

4. No need to show two volcano plots in Fig. 3B.

Response：

Thank you for the helpful comment. According to your comment, we modified Figure 3B.

5. Include a table of fold change and significance in the supplementary materials.

Response:

Thank you for the important comment. We included a table of fold change and significance in the supplementary materials. In S1 Table, fold change and significance were shown for the 9 proteins selected on the volcano plot.

To Reviewer #2: 

1. Following sentences in the abstract need to be rearranged and refined – “Certain factors……..profiling of tumor stroma.” should be changed to “Certain protein…………...invasion and metastasis.”, and moved after first sentence. “We selected….. and 2015” should be changed to “we selected……and 2015 and subjected to comprehensive proteomic profiling of tumor stroma”.

Response:

Thank you for a valuable comment. We rearranged and refined the abstract according to your suggestions.

2. It would be best to collect stromal areas from different areas within one tumor section since there could be significant stromal heterogeneity.

Response:

Thank you for the comment. We totally agree with you on this point. Actually, we collected stroma from different areas on the same section up to a total of 8 mm2. In the materials and methods section of the previous manuscript, “From each patient sample, a stromal area of 8 mm2 (total) was targeted and selectively microdissected” had been described. We rewrote this sentence as follows. “From each patient sample, a stromal area was targeted and selectively microdissected. Stromal samples were collected from different areas within one tumor section up to a total of 8mm2 (total) (Fig2).

3. How many technical replicates were taken in the proteomic analyses?

Response:

Thank you for the important comment. The results were obtained from triplicate LC-MS/MS runs for each sample. We added “The results were obtained from triplicate LC-MC/MC runs for each sample.” in the methods section. 

4. In IHC analyses, discuss why the identified stromal proteins show minimal stromal staining?

Response:

Thank you for the comment of critical importance. Some proteins identified by proteomics were not stained by immunohistochemistry. This might be due to effects of antibody quality or protein amount. Since candidate proteins were selected by relative comparison between the two groups, the amount of protein might not be enough for immunohistochemistry. We added the sentence in the discussion section as follows. “Some proteins identified by proteomics were not stained by immunohistochemistry. This might be due to effects of antibody quality or protein amount. Since candidate proteins were selected by relative comparison between the two groups, the amount of protein might not be enough for immunohistochemistry.”

5. All figure legends should have been written separately.

Response:

In the manuscript body formatting guidelines of PLOS one, it was written that each figure caption should appear directly after the paragraph in which they are first cited.

---

## [Editor Report · Decision Letter 1]

25 Jun 2020

Stromal expression of hemopexin is associated with lymph-node metastasis in pancreatic ductal adenocarcinoma

PONE-D-20-00932R1

Dear Dr.Takadate,

We’re pleased to inform you that your manuscript has been judged scientifically suitable for publication and will be formally accepted for publication once it meets all outstanding technical requirements.

Kind regards,

Eithne Costello

Academic Editor

PLOS ONE

Additional Editor Comments (optional):

Line 379; Please change the following sentences 

Conversely, Panc-1 migration did not significantly differ from control at each hemopexin

concentration. However, it was 20% higher than control at 1μM hemopexin. There seemed to be a similar tendency to MIA PaCa-2 in the migration.

to

At the highest concentration of hemopexin, a 20% increase in Panc-1 migration was observed, although this did not reach statistical significance. 
---

## [Editor Report · Acceptance letter]

30 Jun 2020

PONE-D-20-00932R1 

Stromal expression of hemopexin is associated with lymph-node metastasis in pancreatic ductal adenocarcinoma 

Dear Dr. Takadate:

I'm pleased to inform you that your manuscript has been deemed suitable for publication in PLOS ONE. Congratulations! Your manuscript is now with our production department. 

Kind regards, 

on behalf of

Dr. Eithne Costello 

Academic Editor

PLOS ONE